# A Circular Argument:
# Does RoPE *need* to be Equivariant for Vision?

## Abstract

Rotary Positional Encodings (RoPE) have emerged as a highly effective technique for one-dimensional sequences in Natural Language Processing spurring recent progress towards generalizing RoPE to higher-dimensional data such as images and videos. The success of RoPE has been thought to be due to its positional equivariance, i.e. its status as a *relative* positional encoding. In this paper, we mathematically show RoPE to be one of the most general solutions for equivariant positional embedding in one-dimensional data. Moreover, we show Mixed RoPE to be the analogously general solution for $M$-dimensional data, if we require commutative generators – a property necessary for RoPE's equivariance. However, we question whether strict equivariance plays a large role in RoPE's performance. We propose Spherical RoPE, a method analogous to Mixed RoPE, but assumes non-commutative generators. Empirically, we find Spherical RoPE to have the equivalent or better learning behavior as its equivariant analogues. This suggests that relative positional embeddings are not as important as is commonly believed for vision. We expect this discovery to facilitate future work in positional encodings for vision that are faster and generalize better by removing the preconception that they must be relative.

## 1 Introduction

Recently, Rotational Positional Embeddings (RoPE) [61] have gained popularity, touting an emphasis on the *relative* position between two tokens rather than their absolute positions [18, 22, 37, 40]. Because attention between tokens only depend on their relative distance, the network has *shift-equivariance*. The most general form of rotary encoding is LieRE [46]. However, when extending to higher dimensions, requires one to either give up shift-equivariance or make constraints on the rotations [46, 76, 53]. In this work, we unify recent extensions of RoPE to $M$-dimensions based on the constraints they make on the generators of LieRE. We show that LieRE is shift-equivariant if and only if it can be decomposed into the simpler Mixed RoPE, or the more popular Axial RoPE if further constrained.

However, while RoPE is often claimed to be successful due to its shift-equivariance, the validity of that claim and necessity of equivariance has not been thoroughly tested. We propose Spherical RoPE which strictly takes the assumption of non-commuting generators – thus, non-equivariant – to test this claim. We find Spherical RoPE to perform as well as Mixed RoPE while strictly outperforming Axial RoPE on vision tasks. We also show that Axial RoPE with a single shared frequency performs significantly worse, despite still being equivariant. Thus, we conclude equivariance does not seem to be the primary contributor for RoPE's success in vision.

Submitted to 39th Conference on Neural Information Processing Systems (NeurIPS 2025). Do not distribute.

## 2  Background

### 2.1  Rotary Positional Encodings (RoPE)

Rather than adding a positional embedding to the patch embedding, RoPE proposed to *modify the queries and keys* by rotating them in pairs.

$$RoPE(\mathbf{z}, p) = \mathbf{R}_p \mathbf{z} = \begin{bmatrix} \mathbf{R}_{p\omega_1} & \mathbf{0} & \cdots \\ \mathbf{0} & \mathbf{R}_{p\omega_2} & \cdots \\ \vdots & \vdots & \ddots \end{bmatrix} \begin{pmatrix} \mathbf{z}_1 \\ \vdots \\ \mathbf{z}_D \end{pmatrix} \tag{1}$$

where $\mathbf{R}_{\omega_d p_t}$ is a rotation matrix, $\omega_i$ is a rotation frequency for the corresponding pair. We use the convention that real-valued queries and keys will have dimension $N$ and the number of sub-vectors (pairs) is dimension $D$.

For images, where positions are two-dimensional, RoPE is often extended with Axial RoPE [56], where each position rotates independent sub-vectors. However, because the horizontal and vertical directions are treated independently, this method struggles with representing oblique attention patterns [22]. One rotate the same pair by both positional coordinates where the amount of rotation caused by each is a parameterized for each pair. This is known as Mixed RoPE [22]. Alternatively, one can take the Lie algebra perspective and learn a skew-symmetric generator matrix for each positional coordinate. This known as LieRE [46] and has the benefit of rotating beyond 2D sub-vectors.

## 3  The Generality of Learned RoPE and Mixed RoPE

While RoPE is proposed by rotating 2D sub-vectors of the querys and keys, LieRE can perform the full $N$D rotation. However, by taking the spectral decomposition of the generators, LieRE can be reparameterized into RoPE. For proofs see Appendix K.

> **Proposition 1.** *Any $D$-dimensional rotation can be parameterized by RoPE with learned frequencies.*

### 3.1  Extending RoPE to more than one dimension

While this proof works for 1D positions, it does not generalize to $M$-D without introducing extra inductive biases or giving up equivariance. By imposing constraints on $\mathcal{A}_x$ and $\mathcal{A}_y$, we can categorize the other RoPE methods based on the assumptions made.

**Generators rotate independent subspaces.**   For example, one can impose the assumption that $p_x$ and $p_y$ rotate independent subspaces in $\mathbb{R}^N$. Mathematically, this assumption would imply that

$$\forall d \in [1, D] : \lambda_d^{(x)} = 0 \text{ or } \lambda_d^{(y)} = 0, \tag{2}$$

where $\lambda_d^{(x)}$ and $\lambda_d^{(y)}$ are the eigenvalues of $\mathcal{A}_x$ and $\mathcal{A}_y$, respectively. This is equivalent to rotating independent components of the query/key as done by Axial-RoPE.

**Commutative generators.**   For LieRE to be equivariant, we only need to ensure that the generators commute. If we make this assumption, then we arrive at Mixed RoPE.

> **Proposition 2.** *Any $M$-dimensional LieRE with commutative generators can be parameterized by Mixed RoPE.*

This means that any of the more recent extensions that assume commutativity in the generating matrices of LieRE such as ComRoPE [76] or STRING [53], ultimately are alternative implementations of Mixed RoPE. However, it is not clear that requiring commutativity is necessary or even beneficial. We propose an ablation to the need of equivariance in the form of Spherical RoPE which strictly removes commutativity of the generators.

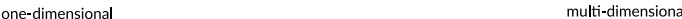

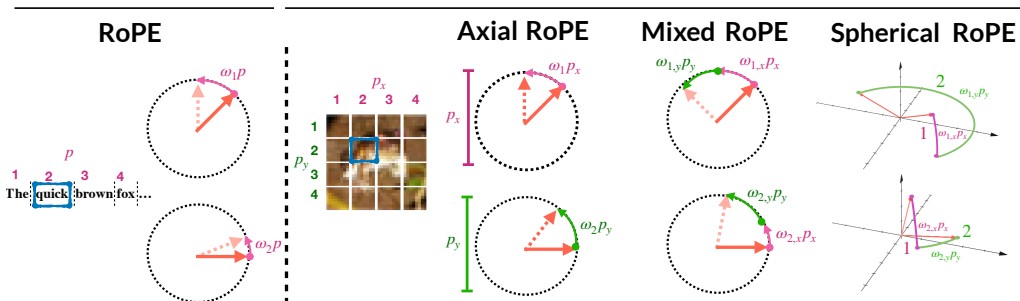

Figure 1: Diagram of each rotary embedding's effect on $\mathbf{z}_d$. While Mixed RoPE effect 2D vector pairs, Spherical RoPE effects 3D vector triplets. Axial RoPE rotates independent dimensions for $p_x$, thus containing *pairs of pairs*, or effectively quadruples. Each $\mathbf{z}$ contains $D$ sub-vectors rotating at different frequencies. While the order in which the rotations are applied does not matter for Axial or Mixed RoPE, order matters for Spherical RoPE. Explicitly, the triplet is first rotated around the axis associated with $p_x$ and then rotated around the axis associated with $p_y$.

## 4 Experiments

When extending RoPE to more than one dimension, we must either constrain ourselves to commuting Lie algebras or give up relativity. We therefore ask the question: Why does RoPE work? Which properties should be preserved for generalizing RoPE to vision? To explore this question, we propose two new RoPE variants: *Spherical RoPE*, which takes an non-commutative assumption, and *Uniform-Frequency RoPE*, which uses a single fixed rotation frequency across all dimensions.

**Spherical RoPE** We propose Spherical RoPE as a method between Mixed RoPE and LieRE that minimally changes 2D RoPE to break equivariance. Spherical RoPE embeds position as

$$\varphi(\mathbf{z}_d, \mathbf{p}) = \mathcal{Y}_{\omega_{dx}x}\mathcal{R}_{\omega_{dy}y}\mathbf{z}_d, \tag{3}$$

where $\mathbf{z}_d \in \mathbb{R}^3$ is now a triplet instead of a pair, and $\mathcal{Y}$ is a block diagonal of $3 \times 3$ *yaw* matrices and $\mathcal{R}$ is a block diagonal of *roll* matrices.

$$\mathcal{Y}_{\omega_{dx}x} = \begin{bmatrix} \cos(\omega_{dx}x) & -\sin(\omega_{dx}x) & 0 \\ \sin(\omega_{dx}x) & \cos(\omega_{dx}x) & 0 \\ 0 & 0 & 1 \end{bmatrix} \qquad \mathcal{R}_{\omega_{dy}y} = \begin{bmatrix} 1 & 0 & 0 \\ 0 & \cos(\omega_{dy}y) & -\sin(\omega_{dy}y) \\ 0 & \sin(\omega_{dy}y) & \cos(\omega_{dy}y) \end{bmatrix}. \tag{4}$$

Intuitively, rather than RoPE rotating around a circle, Spherical RoPE rotates around a sphere using Euler angles. Importantly, spherical rotations like LieRE are *non-commutative* making them *not equivariant*. In fact, their generators are strictly *non-commutative*, $\mathcal{A}_x\mathcal{A}_y \neq \mathcal{A}_y\mathcal{A}_x$. While this does not mean Spherical RoPE is incapable of learning or approximating equivariance throughout the network, it is the component of LieRE removed by Mixed RoPE.

Table 1: Table listing the properties of each of the rotary-based methods.

| Positional Encoding | Vision | Strictly Equivariant | Oblique Directions | Requires Learning |
|---|---|---|---|---|
| Rotary (RoPE) [61] | ✗ | ✓ | N/A | ✗ |
| Axial RoPE [60] | ✓ | ✓ | ✗ | ✗ |
| Mixed RoPE [22] | ✓ | ✓ | ✓ | ✓ |
| LieRE [46] | ✓ | ✗ | ✓ | ✓ |
| Spherical RoPE | ✓ | ✗ | ✓ | ✗ |
| Uniform RoPE | ✓ | ✓ | ✗ | ✗ |

**Uniform-Frequency RoPE.** For an initial evaluation on the impact of relative position, we propose Uniform-Frequency RoPE. For this method, we perform Axial RoPE with a single frequency shared across all rotation matrices. While still being relative, this serves as a more restricted version of RoPE. If this method performs significantly worse than other methods, it indicates more importance of having a range of frequencies than equivariance. We implement uniform frequencies for Axial RoPE to gauge against relative importance of equivariance.

Table 2: Performance comparison (top-1 accuracy) across datasets and methods.

| | Top-1 Accuracy (%) | |
|---|---|---|
| **Fixed Encoding** | **CIFAR100** | **ImageNet** |
| Learned APE | $64.2_{\pm 0.9}$ | 68.8 |
| Axial RoPE | $72.1_{\pm 0.6}$ | 70.7 |
| Uniform RoPE (Our Ablation) | $70.5_{\pm 0.2}$ | 70.0 |
| Spherical RoPE (Our Ablation) | $73.2_{\pm 0.4}$ | **70.9** |
| **Learned Encoding** | | |
| Learned Axial RoPE | $72.9_{\pm 0.6}$ | 70.4 |
| Mixed RoPE | $\mathbf{74.7}_{\pm 0.3}$ | 70.3 |
| Learned Spherical RoPE (Our Ablation) | $74.1_{\pm 0.4}$ | 70.4 |
| LieRE | | 74.2 |

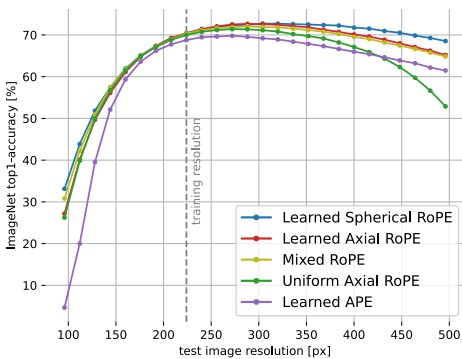

Figure 2: Dependence of accuracy on image resolution for ViT-S with various positional embedding methods on ImageNet1k.

## 5   Results

To evaluate the importance of different properties of positional embeddings in vision transformers, we trained the same ViT with different positional embeddings on CIFAR100 and ImageNet-1K. We start by evaluating the models on images of the same resolution as during training. If equivariance is important, we would see Axial and Mixed RoPE to perform better than Spherical RoPE, which lacks equivariance. On the other hand, if oblique frequencies are important, then we would obeserve Mixed and Spherical RoPE to do better than Axial RoPE, which does not capture oblique directions. We do not find either of the two to be the case: All three methods perform similarly in terms of top-1 accuracy both on CIFAR-100 and ImageNet (Table 1), suggesting that neither equivariance nor capturing oblique directions is important.

The results from training on smaller subsets of the CIFAR100 training data and for the VOC segmentation task can be found in Appendix I. Intuitively, these tasks should favor shift-equivariance since less data favors stronger inductive biases. However, even in these settings Spherical RoPE performs on-par or better.

When comparing to absolute positional encodings, we observe that all forms of RoPE perform better than learned APE (Table 1). This includes Uniform RoPE, the variant that uses only a single frequency. Moreover, all forms of RoPE using diverse frequencies outperform Uniform RoPE and have similar performance (whether they are learned or not), suggesting that diversity of frequencies is important.

Lastly, we asked how well different PEs generalize across image sizes. Equivariance is often thought to aid model generalization. However, when evaluating each model using higher resolutions images, i. e. increasing the number of patches, we found Spherical RoPE to be the most effective method (Fig. 2), suggesting equivariance may not be the reason for RoPE's generalization.

## 6   Conclusion

Because we see very little variation between Spherical RoPE and the equivariant methods, we conclude that equivariance is only a minor contributor to the increased performance seen by RoPE for vision. In fact, Spherical RoPE appeared to extrapolate to higher resolutions better than other methods. This could suggest that oblique frequencies are important for extrapolation. However, Mixed RoPE can also represent oblique directions, but did not outperform Axial RoPE. Thus, neither equivariance nor oblique directions appear to be significant for vision transformers.

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

## A  Broader Impact

This work is fundamental research. While this work could lead to the discovery of better positional encodings and higher performing visual foundation models, the positivity or negativity of this impact is determined by the downstream task and not this work.

## B  Limitations

While our results do not show relative embeddings to be detrimental, we believe them to be evidence that equivariance is not the reason for RoPE's success. However, our experiments were performed in Vision where the number of tokens is limited compared to the long context lengths of NLP. Moreover, the datasets are not what many believe to be "at scale". While Spherical RoPE and LieRE would intuitively favored at scale over Axial RoPE, as they have less inductive bias, it is unclear whether inductive bias and equivariance is favored at scale [8].

It has also been shown that vision is *not* a purely equivariant task and benefits from relaxed equivariance [16]. Our results do not show that equivariance is not useful in tasks that are grounded in physics and obey strict symmetries.

## C  Literature Review

### C.1  Natural Langauge Processing

In natural language, positional encoding has been used to break the permutation, "bag of words", symmetry [65]. Although this could be done by learning a vector per position, this is both memory-expensive for large context sizes making it practical to apply to only the first layer. Moreover, it does not allow for extrapolation at test time to context sizes beyond training. Thus, it is favorable to perform positional embeddings with a predictable deterministic function. One way of doing this is to make the attention relative with local receptive fields, as is done implicitly in convolutional neural networks [10]. Sinusoidal positional embeddings were proposed due to approximate local and shift-invariant properties of Random Fourier Features [51]. Since sinusoidal, other methods have been proposed to get guaranteed shift invariance by explicitly parameterizing based on distance [55, 48, 49]. However, these methods require a positional embedding for every pair of positions which is not supported by many of the efficient attention optimizations such as Flash attention [14] [3].

Rotary Positional Embeddings (RoPE) have become the staple in NLP having recently been adopted by many of the large language models [67, 18, 63, 37, 23]. However, these methods also use causal masking, which has been shown to allow models with no positional embedding to recover absolute position [20, 73, 66, 28]. This has lead to questions on the importance of relative position [4].

In language, there has also been extensions to RoPE proposed through NTKs and kernel methods [9]. However, these methods have not, to our knowledge, seen use in vision.

### C.2  Vision and Video

Vision transformers were introduced in Dosovitskiy et al. [15] and, though they tried sinusoidal position encodings, found learnable position encodings to perform best. For convolution-esque models such as SWin transformers, relative positional encodings have been popular [41, 12]. More recently, RoPE has been shown to be an efficient and simple way to have relative embeddings and has been extended to 2D using Axial and Mixed RoPE. Going beyond 2D to Video data, Axial RoPE has become increasingly popular. The extension was first attributed to Wang et al. [67] as 3D-RoPE or M-RoPE, leading to two separate Video-RoPE papers from Wei et al. [68] and Liu et al. [42]. Both of these focus on the order of the position enumeration and interleaving positions. However, this should not be a problem if frequencies are not deterministic, *or* if frequencies are indexed by both $d$ and modality $m$ as done in Eq **??**. We highly recommend using either Mixed RoPE or LieRE which extend naturally for videos.

LieRE embeddings have thus far been the most general form of RoPE to $N$-D. However, Schenck et al. [53] has claimed the method to have a large memory footprint and proposed STRING. This

paper, a preprint released concurrently with the writing of this manuscript, follows much of the same math as this paper. However, they did not recognize that an orthogonal matrix is implicitly learned by the query and key matrix. Moreover, their method relies on commuting Lie algebras. From our insights in Section 3, their method can likely be viewed as a slower implementation of $N$-D Mixed-RoPE.

It is also worth noting that positional encodings have also been explored within vision through the area of Neural Fields [71]. Traditional coordinate MLPs have been found to be biased toward low-frequency functions [62] leading to more advanced positional encodings such as Random Fourier Features [51] or sinusoidal activation functions [57]. These implicit functions have been used to encode attention and message passing in graph neural networks with recent work being put in to make these functions equivariant to symmetry transformations [52, 7, 31].

### C.3  Graphs and AI in Science

Positional encodings are well studied within graph neural networks [36, 47]. Graphs are limited in their expressivity up to the Weisfeiler-Lehman (WL) graph isomorphism test [72], so positional encodings can break the isomorphism symmetry [21, 74]. Within this community, they propose *spectral attention* and graph Laplacians for positional encoding [33]. These methods seem extremely close to our analysis of RoPE, but from a very different perspective. We show that the frequencies of RoPE can be interpreted as the eigenvalues of an orthogonal transformation by taking the spectral decomposition.

In an overlapping vein, relative position encodings have been studied in terms of equivariant graph neural networks, often for scientific disciplines such as molecular physics [7, 54] or drug discovery [24]. One method to achieve equivariance is through defining relative coordinate frames [32]. This corresponds to the learned relative positional method described in Shaw et al. [55], but can be generalized to higher dimensions and different transformation using bi-invariant distance functions [5, 31, 69]. The message-passing functions of these works correspond to a generalization of attention scores [17].

However, even in these tasks with physics-grounded symmetries, the need for equivariance is hotly debated. While AlphaFold [24] was originally touted as the example of the success of equivariant inductive biases in science, AlphaFold 3 [1] explicitly stated that they benefited from removing this inductive bias at scale. However, while the harm of inductive bias at scale is the prevalent zeitgeist, it is not an established fact [8].

### C.4  Computational Neuroscience

Coupled oscillators have become a growing area of interest within computational neuroscience [29, 30, 59]. By observing the projection of the RoPE circles onto the real axis, one can interpret RoPE as time progression in $D$ uncoupled, undamped harmonic oscillators. This perspective naturally connects RoPE to Löwe et al. [43]'s series of papers on complex autoencoders and their extensions [44, 45].

In another, vein of research, there has been some work in hyper-dimensional computing[25, 26] in Phasor and Residue VSAs [34] which represent concepts as rotations around unit circles in high-dimensional spaces. These representations have strong connections with RoPE. Additionally, progress has been made in hypothesizing how biological neural networks encode positional knowledge with hexagonal grid cells, which can be represented as a discrete sum of three periodic functions oriented at the cubic roots of unity[58].

### C.5  Generality of RoPE

The generality of RoPE has been found by others. Schenck et al. [53], Su [60], and Liu and Zhou [38] all propose proofs similar to Proposition 1. However, Schenck et al. [53] miss that the orthogonal transformation can be incorporated into key matrix. Liu and Zhou [38] and Su [60] take the assumption of *reversibility*, which leads to the independent eigenvalue assumptions of Axial RoPE. All three works take the assumption of an abelian subgroup –ie commutative generators, – but miss the generality of Mixed RoPE. While Su [60] propose quaternions – i. e. spherical rotations – as a direction, they immediately dismiss it as a *no-go* because they lack equivariance. This exemplifies the

439 "circular argument," where equivariance is assumed to be necessary because work will not investigate
440 non-equivariant positional encodings because equivariance is necessary.

441 Because our derivation was found independently of these works and the previous works are, to our
442 knowledge, not published, we have left in Proposition 1. We would like to acknowledge their work,
443 but retain the flow of this paper.

## D  Notation

| Symbol / Term | Dimension | Meaning | Notes |
|---|---|---|---|
| $\mathbf{x}_i$ | $\mathbb{R}^D$ | Patch/token/content vector of token $i$ | Raw input embedding |
| $x_i$ | $\mathcal{X}$ | Abstract content of token $i$ | Raw input embedding |
| $p_i$ | $\mathbb{R}^M$ or $\mathcal{P}$ | Position of token $i$, can be $M$-D or abstract $\mathcal{P}$ | Scalar (1D) or vector (2D) |
| $m$ | $\mathbb{Z}$ | Modality index | e.g., $x$, $y$, time |
| $M$ | $\mathbb{Z}$ | Number, or space, of Modalities | |
| $D$ | $\mathbb{Z}$ | Hidden dimension | Number of pairs/triples/quadruples |
| $T$ | $\mathbb{Z}$ | Number of Tokens | |
| $\mathbf{W}_q, \mathbf{W}_k, \mathbf{W}_v$ | $\mathbb{R}^{\mathcal{X} \times D}$ | Query, Key, Value Matrices | |
| $\mathbf{q}$ | $\mathbb{R}^N$ | $\mathbf{q}_i = \mathbf{W}_q x_i$ | Query vector |
| $\mathbf{k}$ | $\mathbb{R}^N$ | $\mathbf{k}_j = \mathbf{W}_k x_j$ | Key vector |
| $\mathbf{v}$ | $\mathbb{R}^N$ | $\mathbf{v}_j = \mathbf{W}_v x_j$ | Value vector |
| $\mathbf{Q}, \mathbf{K}, \mathbf{V}$ | $\mathbb{R}^{T \times N}$ | Query, Key, Values | $T$ tokens, $D$ latent dimensions |
| $\varphi(x, p)$ | $\mathcal{X} \times \mathcal{P} \to \mathbb{R}^D$ | Positional Encoding function | |
| $\mathbf{Z}$ | $\mathbb{R}^{T \times N}$ | Output of Attention | $\mathbf{Z} = \text{Attention}(\mathbf{Q}, \mathbf{K}, \mathbf{V})$ |
| $a(i, j)$ | $\mathbb{R}$ | Attention weight | Softmax of attention scores |
| $\alpha(\mathbf{q}, \mathbf{k})$ | $\mathbb{R}$ | Attention score | Inner product $\mathbf{q}^\top \mathbf{k}$ |
| $\omega_d / \lambda_d$ | $\mathbb{R}$ | Rotation frequency for dimension $d$ | Equivalent to eigenvalue of generator |
| $\mathbf{q}_d$ | $\mathbb{R}^{2/3/4}$ | Query pair/triple/quadruple at dimension $d$ | After RoPE or LieRE applied |
| $\mathbf{R}_{\omega_d p}$ | $\mathbb{R}^{2 \times 2}$ | $2 \times 2$ rotation matrix | Rotation based on frequency and position |

Table 3: Summary of Notations and Key Concepts

| Positional Encoding | Vision | Learned | Extrapolation | QK Separable | Relative | Linear Flow | Used In |
|---|---|---|---|---|---|---|---|
| Absolute (Sinusoidal) | ✗ | ✓/✗ | ✓ | ✓ | ✓ | ✗ | Transformer[65] |
| Absolute (Learned) | ✓ | ✓ | ✗ | ✓ | ✓ | ✗ | BERT, GPT, ViT[15] |
| Absolute (Random-Fourier) | ✗ | ✗ | ✓ | ✓ | ✗ | ✓ | FNet[35], Performer [11] |
| Relative (Learned) | ✗ | ✓ | ✗ | ✗ | ✗ | ✗ | Transformer-XL, T5 [50] |
| ALiBi | ✗ | ✓/✗ | ✓ | ✓ | ✓ | ✓ | LLaMA 2 [18], ALiBi [49] |
| NoPE | ✗* | ✗ | ✓* | ✓* | ✓* | ✓* | LLaMA 4 [2] |
| Rotary (RoPE) | ✗ | ✗ | ✓ | ✓ | ✓ | ✓ | Contemporary LLMs [67, 18, 63, 23] |
| Axial RoPE | ✓ | ✓/✗ | ✓ | ✓ | ✓ | ✓ | VisionLLaMA[13], Qwen2[67], VideoRoPE[68] |
| Mixed RoPE | ✓ | ✓ | ✓ | ✓ | ✓ | ✓ | Heo et al. [22] |
| LieRE | ✓ | ✓ | ✓ | ✓ | ✗ | ✓ | [46] |
| Spherical RoPE | ✓ | ✓/✗ | ✓ | ✓ | ✗ | ✓ | Ours |
| Uniform RoPE | ✓ | ✓/✗ | ✓ | ✓ | ✓ | ✓ | Ours |

Table 4: Comparison of positional encoding methods in transformer models. *NoPE makes some properties trivially true.

# E   Positional Encoding Properties

Rotary positional embeddings were derived in Su et al. [61] by drawing equations from assumed properties. While these appear as arithmetic assumptions and equations in their work, we formalize what properties these assumptions imply and why we may choose these assumptions in this section. In their paper, to derive their equations, they use equivariance (relativity), query-key separability of the positional encoding, linearity and incompressability, locality, and query-key symmetry.

1. Equivariance/Relativity: Attention score should be affected only by the relative position of two tokens, i. e. have the form

$$\alpha(x_i, x_j, p_i, p_j) = \hat{\alpha}(x_i, x_j, p_i - p_j).$$ (5)

2. Key-query seperability: The positional encoding, $\varphi$, of the query should not depend on the position of the key

$$\alpha(x_i, x_j, p_i, p_j) = \bar{\alpha}(\varphi(x_i, p_i), \varphi(x_j, p_j))$$ (6)

3. Linearity: The positional encoding should be a linear flow, see Appendix E.3. Namely,

$$\varphi(\varphi(x, p_i), p_j) = \varphi(x, p_i + p_j).$$ (7)

4. Locality: The attention score between two tokens should decay with distance

$$\lim_{|p_i - p_j| \to \infty} \alpha(x_i, x_j, p_i, p_j) = 0$$ (8)

## E.1   Relativity and Equivariant

We use the term *equivariant* interchangably with *relative*. Strictly speaking, one should specify the transformation or group you would like to be relative to, e. g. shift/rotation or $SO(2)$. As previous literature always refers to relative positional bias in terms of shifts/translations, in the main text, this is what we mean. We use the term equivariance to be the generalization of relativity beyond language because we would like to refrain from using the term "relativity" to describe the property of being a relative PE too often due to its connotation within theoretical physics. First, we define relative in the case of positional encodings in language as

$$\alpha(x_i, x_j, p_i, p_j) = \hat{\alpha}(x_i, x_j, p_i - p_j).$$ (9)

In the rest of this section, we mathematically explore where this equation comes from.

The behavior we are trying to capture is that if we renumber the words in the sentence, it should not affect the attentions score. Intuitively, if a text is padded with spaces at the beginning, that will not have a significant effect on the meaning of the sentences. We can ensure this by colloquially saying that the attention between two words should depend on the distance between them. Notice, that strictly speaking this is not a proper distance, since it can be negative; it is, instead, a *signed* distance function. Though this may seem pedantic in one dimension, in two dimensions defining a distance function is less unique. For example, one may choose $\mathbb{L}_1$ or $\mathbb{L}_2$ distance metrics. Because distance functions are more nebulous, it makes more sense to define relative in terms of the transformations that we would like our attention score to be independent of.

$$\alpha(x_i, x_j, p_i, p_j) = \alpha(x_i, x_j, T(p_i), T(p_j)).$$ (10)

These transformations can be combined to generate a set of transformations which leave the attention score unchanged, or *symmetric*. This set has the mathematical properties of a group and is known as a symmetry group. We can index transformations by elements in the symmetry group, $g \in G$, and let the elements act on

$$\alpha(x_i, x_j, p_i, p_j) = \alpha(x_i, x_j, g.p_i, g.p_j). \tag{11}$$

As an example, $g$ could represent an angle, $\theta$, and it may act on a vector $\mathbf{p}$ as a rotation $g.\mathbf{p} = \mathbf{R}_\theta \mathbf{p}$.

Connecting everything back to Eq. 9, Noether's theorem states that any continuous symmetry can be expressed as a conservation law. This allows us to introduce bi-invariant function [31, 69], or "Noether charge", $\beta(p_i, p_j)$, that is invariant under the group action,

$$\beta(p_i, p_j) = \beta(g.p_i, g.p_j) \implies \beta(p_i, p_j) - \beta(g.p_i, g.p_j) = 0. \tag{12}$$

Thus, we can express our symmetry group through isodistances of $\beta$,

$$\alpha(x_i, x_j p_i, p_j) := \hat{\alpha}(x_i, x_j, \beta(p_i, p_j)). \tag{13}$$

For example, we can pick the function

$$\beta(p_i, p_j) = p_i - p_j = (p_i - p_0) - (p_j - p_0) = \beta(p_i - o, p_j - p_0) \tag{14}$$

If we were to define $\beta(p_i, p_j) = |p_i - p_j|$, then we we would additionally be equivariant to reflection of the order of tokens in a sentence. If we trivially define $\beta(p_i, p_j) = C$, then we arrive at bag of words, or no positional encoding (NoPE). For a list of common transformations and their corresponding bi-invariants see Theorem 1 of Bekkers et al. [5].

## E.2 Query-Key Separability

Query and key separability is important for efficiency reasons. If we can decompose our positional encoded attention score as,

$$\alpha(x_i, x_j, p_i, p_j) = \alpha(\varphi(x_i, p_i), \varphi(x_j, p_j)) \tag{15}$$

then we can pre-compute the positional encoding for the queries and keys on time making the computation $O(T)$. If the positional encoding is not separable, then it will need to be computed for *every pair*, $(i, j)$[41, 50, 55]. Although there are many symmetries that can be exploited to make this not a quadratic computation, it removes the symmetries exploited by efficient attention mechanisms [6, 11, 27].

## E.3 Linear Flow Property

The property of being a "flow" was first proposed in Liu et al. [39], however it is not often discussed. It is a property inherently present in RoPE[61], LieRE[46] and ALiBi [49] embeddings, specifically as a *linear flow*.

We use the term *linear flow* for this property because the embedding can be found by repeated application of a linear function. However, the term "linear" this is a small misnomer because it is only *locally* linear. We define a *flow* as function

$$\varphi : \mathbb{R}^N \times \mathbb{R} \to \mathbb{R}^N \tag{16}$$

such that for all $x \in X$ and $p_1, p_2 \in \mathbb{R}$, the following conditions hold:

1. Initial condition (identity at time zero):

$$\varphi(0, x) = x \tag{17}$$

2. Group property (flow property):

$$\varphi(\varphi(\mathbf{x}, p_1), p_2)) = \varphi(x, p_1 + p_2) \tag{18}$$

3. Continuity (or differentiability): $\varphi$ is continuous with respect to its variables, depending on the context

Strictly speaking, continuity is not necessary for positional encodings as positions tend to be integer values. What we really wish to capture with this property is for the positional encoding to be recursively defined. It may be strange to wish to apply the positional encoding multiple times; however, by having the positional encoding as an endomorphism it can allow for more predictable behavior when extrapolating to larger contexts, which we suspect helps the model train.

We define a position embedding to be a *linear flow* if the flow has the form:

$$\varphi(\mathbf{x}, \Delta p) = \mathbf{A}\mathbf{x}, \tag{19}$$

for $\mathbf{A} \in \mathbb{R}^{N \times N}$ and $\mathbf{x} \in \mathbb{R}^N$, where $\Delta p$ is the increment rate for position. By Eq. 18, any position $p := p_0 \Delta p$ can then be attained by,

$$\varphi(\mathbf{x}, p) = \mathbf{A}^{p_0}\mathbf{x}. \tag{20}$$

This can be seen as a *geometric series* if $\mathbf{A}$ is a scalar as seen in Press et al. [49]. If we let $\Delta t$ become infinitesimal, then we can express the recurrence relationship as the ODE,

$$\frac{\partial \varphi}{\partial t} = \mathcal{A}\varphi \tag{21}$$

which we can integrate to get,

$$\varphi(\mathbf{x}, p) = \exp(\mathcal{A}p)\mathbf{x} \tag{22}$$

This $\mathcal{A}$ is our *generator* of the flow, which is also a generator for a *matrix Lie algebra*, which we focus on in the main text. The matrix exponential, $\exp : \mathbb{R}^{N \times N} \to \mathbb{R}^{N \times N}$, can be unstable for long contexts; similar to the scalar exponential function $e^{xp}$, the function can quickly become large for high values of $x$. However, this can be stable value $x = 0$, since it always results in one. Similarly, the matrix exponential can be stable if the divergence of the flow – trace of the generator – is zero. We call flow "incompressible" or "divergence-free" if the trace of $\mathcal{A}$ is zero, making the determinant of $\mathbf{A}$ unit. If fluid dynamics, this is called *incompressibility*. For fluids, this implies that the flow conserves mass.

If there are more than one generator of the Lie group, $\mathcal{A}_1$ and $\mathcal{A}_2$, then Eq. 18 must be modified to,

$$\varphi(\varphi(\mathbf{x}, \mathbf{p_1}), \mathbf{p_2}) = \varphi(\mathbf{x}, \mathbf{p_1} \circ \mathbf{p_2}), \tag{23}$$

where $\circ$ is the group product. By the Baker–Campbell–Hausdorff formula, $\exp \mathcal{A}_1 p_1 \exp \mathcal{A}_2 p_2 = \exp \mathcal{A}_1 p_1 + \mathcal{A}_2 p_2$ iff the commutator of $\mathcal{A}_1 p_1$ and $\mathcal{A}_2 p_2$ is zero, i. e. the matrices commute. If they do commute, then

$$\varphi(\varphi(\mathbf{x}, \mathbf{p_1}), \mathbf{p_2}) = \varphi(\varphi(\mathbf{x}, \mathbf{p_2}), \mathbf{p_1}) \implies \varphi(\mathbf{x}, \mathbf{p_1} \circ \mathbf{p_2}) = \varphi(\mathbf{x}, \mathbf{p_2} \circ \mathbf{p_1}) \tag{24}$$

thus making $\circ$ commutative and having the same properties as addition, $\circ := $ "+", and Eq. 18 will hold. In this case, the group/flow is known as an *abelian* Lie group, or *abelian flow*. However, if they do not commute, then $\circ$ will not commute and they are known as *non-abelian*. This also makes the flow *non-integrable*.

## E.4 Locality

Locality is often conflated with relativity. The general idea is that tokens far from each other should be independent of one another – i. e. attention should decay as distance grows. This often motivates the definition

$$\lim_{|p_i - p_j| \to \infty} \alpha(x_i, x_j, p_i, p_j) = 0 \tag{25}$$

for $p_i, p_j \in \mathbb{R}$ and $x_i, x_j \in \mathbb{R}^D$. However, this definition is *both* relative and local. We instead define local as,

$$\lim_{|p_i - p_0| \to \infty} \alpha(x_i, x_j, p_i, p_0) = 0. \tag{26}$$

The difference being that $p_0$ is the *origin* position. If an embedding is relative, then the origin is arbitrary and can be defined as $p_i$ or $p_j$. In Press et al. [49], they define the origin vector as the next word. However, they can only do this because of the causal mask.

In general, the most natural way to measure locality is through the concept of the quantum mechanical concept of the *variance of an operator*. We will simply use exponential decay, but we point interested readers to Chapter 3 of Griffiths [19]. This formalism works for RoPE as it is a linear transformation and the attention mechanism defines a Hilbert space.

To be clear, RoPE and LieRE are *not* relative embeddings. This was shown for RoPE in Barbero et al. [4]. Because they are orthogonal matrices, they have unit determinant, which naturally precludes locality.

### E.5  Other properties

For completeness, there are two additional assumptions that are common.

**Adjoint symmetry of the Positional Encoding**   We implicitly assume that the positional encoding is symmetric for the query and key. That is, we assume that the query and key are from the same domain, so the positional encoding has the same representation. More generally, the positional encoding can act differently on the query and key,

$$\alpha(\bar{\varphi}(x_i, p_i), \varphi(x_j, p_j)) = \alpha(\varphi(x_i, p_i), \varphi(x_j, p_j)), \tag{27}$$

where $\bar{\varphi}$ is the positional encoding function for queries. More generally, we can have a relative embedding by letting $\bar{\varphi}$ act on queries differently from the keys. For example, if we let

$$\varphi(x, p) = \exp(\Lambda p) \qquad\qquad \bar{\varphi}(x, p) = \exp(-\Lambda p), \tag{28}$$

where $\Lambda$ is a diagonal matrix. We end up with,

$$\alpha(\bar{\varphi}(x_i, p_i), \varphi(x_j, p_j)) = \mathbf{q}_i^{\top} \exp(\Lambda(p_j - p_i))\mathbf{k}_j, \tag{29}$$

where RoPE can be interpreted as a simple harmonic oscillator, by weakening the symmetry requirement, one could incorporate damping. This can also be used to incorporate graph Laplacian positional encodings into the framework.

**Reversibility**   Reversibility means that the positional encoding is an injective map – that is, every coordinate is mapped to a unique rotation, thus position can be recovered. This property is important in Liu and Zhou [38] and Su [60] to derive Axial RoPE. While it prevents Eq. **??**, it is necessary only for the $D = 1$ case. More generally, Mixed RoPE can learn an injective map for large $D$. Moreover, while having a "lossless" positional encoding is nice mathematically, its practical utility has yet to be soundly justified, especially if the positional encoding is learnable.

## F  Fast Implementation

We follow a vectorized implementation for Spherical RoPE similar to the "fast implementation" proposed in Su et al. [61].

First, apply the rotation directly on after the other:

$$z_d[1] = \cos(\omega_y p_y)\, z_d[1] - \sin(\omega_y p_y)\, z_d[3] \tag{30}$$

$$z_d[3] = \sin(\omega_y y)\, z_d[1] + \cos(\omega_y)\, z_d[3], \tag{31}$$

then

$$z_d[2] = \cos(\omega_y p_x)\, z_d[2] - \sin(\omega_x p_x)\, z_i[3] \tag{32}$$

$$z_d[3] = \sin(\omega_x p_x)\, z_d[2] + \cos(\omega_x p_x)\, z_d[3], \tag{33}$$

where steps 30 and 31 happen simultaneously, and steps 32 and 33 occur at the same time.

## G  Experimental Setup

**Models**    We use the ViT-S backbone from the timm library [70]. The network always has a depth of 12. We keep $N$ as close to constant across models as we can. For CIFAR100, the embedding dimensions are changed from $64 \times N_{\text{heads}}$ to $60 \times N_{\text{heads}}$ to be compatible with pairs, triplets and quadruples. For ImageNet, we make the embedding dimension $63 \times N_{\text{heads}}$ for Spherical RoPE and $64 \times N_{\text{heads}}$ for other methods. For classification, we use a class token to pool the tokens and predict. Unlike the patch tokens, the class token is not affected by any positional encoding.

**CIFAR100**    All experiments on CIFAR100 were performed on one A100 GPUs with a batch size 256. We use a patch size of $4 \times 4$ on the original image size $32 \times 32$. The training uses heavy regularization and augmentations including dropout, MixUp [78] and CutMix [77]. The models are trained for 400 epochs, taking $\sim 40$ seconds per training loop.

**ImageNet**    All experiments on ImageNet1k were performed on four A100 GPUs with a batch size 256. We used cosine learning rate with a learning rate of $3e-3$ for 200 epochs with 5 epochs of linear warm-up. We used a patch size of $16 \times 16$ on the cropped and resized $224 \times 224$ image after applying 3-Augment [64]. We use the LAMB [75] optimizer. All experiments took $\sim 20$ hrs with $\sim 5$ to 8 minutes to complete a training loop depending on method.

**Positional Encodings**    For testing with different resolutions, the images from ImageNet's validation set were normalized, resized and cropped. On training, the patches were assigned position $[-\pi, \pi]$ and for evaluation, the patch positions were extrapolated to the range $[-\frac{P}{P_0}\pi, \frac{P}{P_0}\pi]$. For Learned APE, the positional embeddings are instead interpolated. The fixed frequencies were given by $\omega_d = 1/100^{2d/D}$, where $d$ is the index of the pair/tuple/quadruple. One frequency is shared between both $x$ and $y$ in our implementation of Axial RoPE .

 # H  Hyperparameters

Table 5: Hyperparameters for ImageNet-1K Training

| Category | Setting |
| --- | --- |
| **Model Architecture** | |
| Patch Size | 16x16 |
| Heads | 6 |
| Latent Dimension | 64 (63 for Spherical) $\times$ Heads |
| Depth | 12 |
| Pooling | [CLS] |
| Stochastic Depth | No |
| Dropout | No |
| LayerScale | 1 |
| **Optimization** | |
| Optimizer | LAMB [75] |
| Base Learning Rate | 4e-3 |
| Weight Decay | 0.05 |
| Learning Rate Schedule | Cosine Decay |
| Warmup Schedule | Linear |
| Warmup Epochs | 5 |
| Epochs | 200 |
| Batch Size | 512 |
| Gradient Clipping | ✓ |
| **Precision and Backend** | |
| Precision | Mixed (bfloat16) |
| Backend | torch.autocast |
| **Data Augmentation - Train** | |
| Crop | RandomResizedCrop (192→224) |
| Flip | ✓ |
| 3-Augment | ✓ |
| Color Jitter | (0.3, 0.3, 0.3, 0.0) |
| Mixup [78] | ✗ |
| Cutmix [77] | ✗ |
| Normalization | ImageNet-1K Statistics |
| **Data Augmentation - Test** | |
| Resize | Resize → Resolution |
| Crop | CenterCrop |
| Normalize | ImageNet-1K Statistics |

Table 6: Hyperparameters for CIFAR100 Training

| Category | Setting |
| --- | --- |
| **Model Architecture** | |
| Patch Size | 16x16 |
| Heads | 12 |
| Latent Dimension | $60 \times$ Heads |
| Depth | 12 |
| Pooling | [CLS] |
| Stochastic Depth | 0.1 |
| Dropout | 0.1 |
| LayerScale | ✓ |
| **Optimization** | |
| Optimizer | LAMB [75] |
| Base Learning Rate | 4e-3 |
| Weight Decay | 0.05 |
| Learning Rate Schedule | Cosine Decay |
| Warmup Schedule | Linear |
| Warmup Epochs | 5 |
| Epochs | 400 |
| Batch Size | 1024 |
| Gradient Clipping | ✓ |
| **Precision and Backend** | |
| Precision | Mixed (bfloat16) |
| Backend | torch.autocast |
| **Data Augmentation - Train** | |
| Crop | RandomResizedCrop (32) |
| Flip | ✓ |
| 3-Augment | ✓ |
| Color Jitter | (0.3, 0.3, 0.3, 0.0) |
| Mixup [78] | 0.8 |
| Cutmix [77] | 1.0 |
| Normalization | CIFAR Statistics |
| **Data Augmentation - Test** | |
| Normalize | CIFAR Statistics |

# I  Additional Evaluations

In this section, we include extra evaluations including, basic data scaling, segmentation and speed. We also include additional experiments on the effect of rotation frequencies on Uniform RoPE.

## I.1  Data Scaling

Below we evaluate the data scaling of each method. We partition the CIFAR100

Table 7: Performance on different portions of CIFAR100.

| Dataset Size | Spherical (Learned) | Axial (Learned) | Mixed | Uniform | APE |
|---|---|---|---|---|---|
| 0.2 | 56.04 (**57.2**) | 55.3 (56.6) | 56.9 | 52.82 | 45.9 |
| 0.4 | 63.6 (**65.34**) | 63.3 (62.5) | 64.4 | 59.7 | 53.4 |
| 0.6 | 67.6 (69.8) | 66.0 (66.78) | **70.0** | 64.1 | 57.7 |
| 0.8 | 69.8 (**72.6**) | 69.9 (69.1) | 71.6 | 65.8 | 59.0 |

Equivariance, in theory, should provide better scaling due to its inductive bias. However, we observe that learned Spherical RoPE performs on-par or better than Mixed RoPE with less parameters.

# J  Segmentation

Table 8: Segmentation results (IoU) on VOC with and without augmentation.

| | Spherical | Axial (Learned) | Mixed | Uniform |
|---|---|---|---|---|
| VOC (No Aug.) | **0.45** | 0.42 (0.43) | 0.41 | 0.41 |
| VOC (Simple Aug.) | **0.50** | 0.46 (0.47) | **0.50** | 0.45 |

# K  Proofs and Lemmas

**Axial RoPE Separability**

> **Proposition 3.** *Axial RoPE is separable in $x$ and $y$, that is, the attention score can be decomposed into,*
> $$\alpha(\mathbf{x}_i, \mathbf{x}_j, \mathbf{p}_i, \mathbf{p}_j) = \alpha_{ij}^{(x)} + \alpha_{ij}^{(y)}$$

**Proof.** Suppose we define the dot-product attention score as
$$\alpha(\mathbf{q}, \mathbf{k}) = \mathbf{q}^\top \mathbf{k}.$$

We incorporate *Axial Rotary Positional Embeddings* by rotating each 2-dimensional subvector of the query (and likewise the key). Concretely, if the hidden dimension is $2n$, we partition

$$\mathbf{q} = \begin{bmatrix} \mathbf{q}_{x,1}, \mathbf{q}_{y,1}, \ldots, \mathbf{q}_{x,n}, \mathbf{q}_{y,n} \end{bmatrix}^\top, \quad \mathbf{k} = \begin{bmatrix} \mathbf{k}_{x,1}, \mathbf{k}_{y,1}, \ldots, \mathbf{k}_{x,n}, \mathbf{k}_{y,n} \end{bmatrix}^\top, \quad (34)$$ where each $\mathbf{q}_{x,d}, \mathbf{q}_{y,d}, \mathbf{k}_{x,d}, \mathbf{k}_{y,d} \in \mathbb{R}^2$. At spatial location $\mathbf{p} = (p_x, p_y)$, we apply rotations

$$\mathbf{q}'_{x,d} = \mathbf{R}(\omega_d\, p_x)\, \mathbf{q}_{x,d}, \quad \mathbf{q}'_{y,d} = \mathbf{R}(\omega_d\, p_y)\, \mathbf{q}_{y,d},$$

and similarly for $\mathbf{k}$. Here $\mathbf{R}(\theta) \in \mathbb{R}^{2\times2}$ is the planar rotation by angle $\theta$.
For tokens at positions $\mathbf{p}_i = (p_{i,x}, p_{i,y})$ and $\mathbf{p}_j = (p_{j,x}, p_{j,y})$, their rotated queries and keys yield

$$\alpha_{ij} = \sum_{d=1}^n \Big[ (\mathbf{q}_{x,d})^\top \mathbf{R}\big(\omega_d\, (p_{j,x} - p_{i,x})\big)\, \mathbf{k}_{x,d} + (\mathbf{q}_{y,d})^\top \mathbf{R}\big(\omega_d\, (p_{j,y} - p_{i,y})\big)\, \mathbf{k}_{y,d} \Big].$$

Define the horizontal and vertical components by

$$\alpha_{ij}^{(x)} := \sum_{d=1}^n (\mathbf{q}_{x,d})^\top \mathbf{R}\big(\omega_d\, (p_{j,x} - p_{i,x})\big)\, \mathbf{k}_{x,d}, \quad \alpha_{ij}^{(y)} := \sum_{d=1}^n (\mathbf{q}_{y,d})^\top \mathbf{R}\big(\omega_d\, (p_{j,y} - p_{i,y})\big)\, \mathbf{k}_{y,d}.$$

Hence the total attention decomposes additively:

$$\alpha_{ij} = \alpha_{ij}^{(x)} + \alpha_{ij}^{(y)},$$

demonstrating that *axial* rotary embeddings factorize the positional dependence along each axis. $\square$

**Matrix Exponentiation**  Computing the matrix exponential by exponentiating the eigenvalues is a common result in linear algebra and numerics, however we provide it here for those unfamiliar.

> **Lemma 1.** *Let $\mathbf{A}$ be a diagonalizable matrix $\mathbf{A} = \mathbf{U}\mathbf{\Lambda}\mathbf{U}^{-1}$, then the matrix exponential of $\mathbf{A}$ is given by*
> $$\exp(\mathbf{A}) = \mathbf{U}\exp(\mathbf{\Lambda})\,\mathbf{U}^{-1}$$

**Proof.**
Recall the power-series definition of the matrix exponential:

$$\exp(\mathbf{A}) = \sum_{k=0}^\infty \frac{1}{k!}\,\mathbf{A}^k. \tag{35}$$

Since $\mathbf{A}$ is diagonalizable,
$$\mathbf{A}^k = \big(\mathbf{U}\,\mathbf{\Lambda}\,\mathbf{U}^{-1}\big)^k = \mathbf{U}\,\mathbf{\Lambda}^k\,\mathbf{U}^{-1}. \tag{36}$$

Substituting into the series gives

$$\exp(\mathbf{A}) = \sum_{k=0}^\infty \frac{1}{k!}\big(\mathbf{U}\,\mathbf{\Lambda}^k\,\mathbf{U}^{-1}\big) = \mathbf{U}\Big(\sum_{k=0}^\infty \frac{1}{k!}\,\mathbf{\Lambda}^k\Big)\mathbf{U}^{-1}. \tag{37}$$

628 Because $\mathbf{\Lambda}$ is diagonal, the series $\sum_{k=0}^{\infty} \frac{1}{k!} \mathbf{\Lambda}^k$ is itself the diagonal matrix of scalar exponentials,

$$\exp(\mathbf{\Lambda}) = \mathrm{diag}(e^{\lambda_1}, \ldots, e^{\lambda_n}). \tag{38}$$

629 Hence is well defined, and

$$\exp(\mathbf{A}) = \mathbf{U} \exp(\mathbf{\Lambda}) \mathbf{U}^{-1}. \tag{39}$$

630 $\qquad\qquad\qquad\qquad\qquad\qquad\qquad\qquad\qquad\qquad\qquad\qquad\qquad\qquad\qquad\qquad\square$

631 **Simultaneous-Diagonalizability**    The proof that two (diagonalizable) matrixes are simultaneous-
632 diagonalizability if and only if they are commutative is also a standard result. However, we once
633 again provide it here:

> **Lemma 2.** *Let $\mathcal{A}_x$ and $\mathcal{A}_y$ be skew-symmetric. Then $\mathcal{A}_x$ and $\mathcal{A}_y$ are simultaneously diagonalizable if and only if $\mathcal{A}_x \mathcal{A}_y = \mathcal{A}_y \mathcal{A}_x$ .*

634 **Proof.**
635 Suppose $\mathcal{A}_x$ and $\mathcal{A}_y$ are simultaneously diagonalizable. Then, because they are skew-symmetric,
636 there exists a unitary matrix $\mathbf{U}$ such that

$$\mathbf{U}\Lambda_x \mathbf{U}^\top = \mathcal{A}_x \quad \text{and} \quad \mathbf{U}\Lambda_y \mathbf{U}^\top = \mathcal{A}_y, \tag{40}$$

637 where $\Lambda_x$ and $\Lambda_y$ are diagonal matrices.
638 Then,

$$\mathcal{A}_x \mathcal{A}_y = \mathbf{U}\Lambda_x \mathbf{U}^\top \mathbf{U}\Lambda_y \mathbf{U}^\top = \mathbf{U}\Lambda_x \Lambda_y \mathbf{U}^\top = \mathbf{U}\Lambda_y \Lambda_x \mathbf{U}^\top = \mathcal{A}_x \mathcal{A}_y \tag{41}$$

639 Hence, $\mathcal{A}_x$ and $\mathcal{A}_y$ commute.
640 Now suppose $\mathcal{A}_x$ and $\mathcal{A}_y$ commute, $\mathcal{A}_x \mathcal{A}_y = \mathcal{A}_y \mathcal{A}_x$. Since $\mathcal{A}_x$ and $\mathcal{A}_y$ are skew-symmetric, they
641 are diagonalizable in $\mathbb{C}^{DxD}$, thus there exists a basis of eigenvectors of $\mathcal{A}_x$. Because $\mathcal{A}_y$ commutes
642 with $\mathcal{A}_x$, the eigenspaces of $\mathcal{A}_x$ are invariant under $\mathcal{A}_y$. That is, for any eigenvalue $\lambda$ of $\mathcal{A}_x$, the
643 corresponding eigenspace

$$E_\lambda = \{v \in \mathbb{C}^D : \mathcal{A}_x v = \lambda v\} \tag{42}$$

644 is $\mathcal{A}_y$-invariant: if $v \in E_\lambda$, then

$$\mathcal{A}_x(\mathcal{A}_y v) = \mathcal{A}_y(\mathcal{A}_x v) = \mathcal{A}_y(\lambda v) = \lambda \mathcal{A}_y v \Rightarrow \mathcal{A}_y v \in E_\lambda. \tag{43}$$

645 Now, restrict $\mathcal{A}_x$ to each eigenspace $E_\lambda$. Since $\mathbb{C}$ is algebraically closed and $\mathcal{A}_y|_{E_\lambda}$ is a linear
646 operator on a finite-dimensional space, $\mathcal{A}_y$ is diagonalizable on $E_\lambda$. Thus, we can choose a basis of
647 eigenvectors for $\mathcal{A}_y$ in each $E_\lambda$.
648 Putting these together, we get a basis for $\mathbb{C}^N$ consisting of vectors that are eigenvectors for both $\mathcal{A}_x$
649 and $\mathcal{A}_y$. Therefore, $\mathcal{A}_x$ and $\mathcal{A}_y$ are simultaneously diagonalizable.
650 $\qquad\qquad\qquad\qquad\qquad\qquad\qquad\qquad\qquad\qquad\qquad\qquad\qquad\qquad\qquad\qquad\square$

651 **1-D LieRE is equivalent to RoPE**    In this section, we will more formally prove that the traditional
652 RoPE with learned rotation frequencies is equivalent to 1-D RoPE as proposed in Section 3.

> **Proposition 1.** *Any $D$-dimensional rotation can be parameterized by RoPE with learned frequencies.*

653 **Proof.**
654 We define a rotation to be an orthogonal matrix with positive determinant; that is, it is an element
655 of $\mathbf{R} \in \mathrm{SO}(N)$. We can write any element of $\mathrm{SO}(N)$ via the exponential map $\mathbf{R} = e^{\mathcal{A}}$ where
656 $\mathcal{A} \in \mathfrak{so}(N)$, i.e. $\mathcal{A}$ is a skew-symmetric matrix. It is well-known that the eigenvalues of a real, skew-
657 symmetric matrix are purely imaginary (or zero), and such a matrix is unitarily (i.e. orthogonally)
658 diagonalizable over $\mathbb{C}$, resulting in a spectral decomposition with a purely imaginary eigenvalue
659 matrix. Thus,

$$\mathcal{A} = \mathbf{U}\mathbf{\Lambda}i\mathbf{U}^\dagger \tag{44}$$

660 and, by Lemma 1,

$$\exp(\mathcal{A}) = \mathbf{U} \exp(\mathbf{\Lambda}i) \mathbf{U}^\dagger. \tag{45}$$

661 where, because $\mathbf{\Lambda}$ is diagonal, $\exp(\mathbf{\Lambda})$ is simply the scalar-exponential of each element. The positional
662 encoding of a token to a query can be written as,

$$\varphi(\mathbf{x}, p) = \exp(\mathcal{A}p)\mathbf{W}_q\mathbf{x} = \mathbf{U} \exp(\mathbf{\Lambda}i\, p)\mathbf{W}_{\mathbf{q}}'\mathbf{x} \tag{46}$$

where $\mathbf{W}'_q = \mathbf{W}_q\mathbf{U}$. We assume the same encoding for the key with a different matrix, $\mathbf{W}'_k$ and the same generator, $\mathcal{A}$. This equation can be rewritten as $\varphi(\mathbf{x}, p) = \mathbf{U}RoPE(\mathbf{x}, p)$ by Eq.1. If the attention score is given by $\alpha(\mathbf{q}, \mathbf{k}) = \mathbf{q}^\dagger\mathbf{k}$, where $\dagger$ denotes the Hermitian transpose, then the attention score can be expanded into,

$$\alpha(\mathbf{x}_i, \mathbf{x}_j, p_i, p_j) = RoPE(\mathbf{x}_i, p_i)^\dagger \mathbf{U}^\dagger \mathbf{U} RoPE(\mathbf{x}_j, p_j) \tag{47}$$

$$= RoPE(\mathbf{x}_i, p_i)^\dagger RoPE(\mathbf{x}_j, p_j). \tag{48}$$

Hence, *any LieRE of one generator can be expressed as RoPE with learned rotation frequencies.* $\square$

**Any commutative LieRE is equivalent to Mixed RoPE**   We now prove that multi-dimensional LieRE with commutative generators generalizes directly to Mixed RoPE.

---
**Proposition 2.** *Any $M$-dimensional LieRE with commutative generators can be parameterized by Mixed RoPE.*

---

**Proof.**
Let $\mathcal{A}_1, \dots, \mathcal{A}_M \subset \mathfrak{so}(N)$ be skew-symmetric generators such that $[\mathcal{A}_m, \mathcal{A}_n] = \mathbf{0}$ for all $m, n$. By Lemma 2, commuting normal matrices are simultaneously unitarily diagonalizable. Thus, there exists a unitary $\mathbf{U}$ and diagonal matrices $\mathbf{\Lambda}_1, \dots, \mathbf{\Lambda}_M$ such that

$$\mathcal{A}_m = \mathbf{U}\mathbf{\Lambda}_m i\mathbf{U}^\dagger \quad \text{for all } m = 1, \dots, M. \tag{49}$$

For a position vector $\mathbf{p} = (p_1, \dots, p_M) \in \mathbb{R}^M$, the LieRE positional encoding is

$$\text{LieRE}(\mathbf{x}, \mathbf{p}) = \exp\left(\sum_{m=1}^{M} \mathcal{A}_m p_m\right) \mathbf{W}q\mathbf{x}, \tag{50}$$

which, using Lemmas 1 and 2, can be written as

$$\text{LieRE}(\mathbf{x}, \mathbf{p}) = \mathbf{U} \exp\left(\sum_{m=1}^{M} \mathbf{\Lambda}_m i, p_m\right) \mathbf{U}^\dagger \mathbf{W}_q\mathbf{x}. \tag{51}$$

Let $\mathbf{W}'_q = \mathbf{U}^\dagger\mathbf{W}_q$. Then

$$\text{LieRE}(\mathbf{x}, \mathbf{p}) = \mathbf{U}\text{MixedRoPE}(\mathbf{x}, \mathbf{p}), \tag{52}$$

where MixedRoPE applies elementwise complex rotations

$$e^{i(\lambda_1^{(k)}p_1 + \cdots + \lambda_M^{(k)}p_M)} \tag{53}$$

to each channel $k$, with frequencies $\lambda_m^{(k)}$ learned from $\mathbf{\Lambda}_m$.
If the attention score is given by $\alpha(\mathbf{q}, \mathbf{k}) = \mathbf{q}^\dagger\mathbf{k}$, then

$$\alpha(\mathbf{x}_i, \mathbf{x}_j, \mathbf{p}_i, \mathbf{p}_j) = \text{MixedRoPE}(\mathbf{x}_i, \mathbf{p}_i)^\dagger \mathbf{U}^\dagger \mathbf{U} \text{MixedRoPE}(\mathbf{x}_j, \mathbf{p}_j) \tag{54}$$

$$= \text{MixedRoPE}(\mathbf{x}_i, \mathbf{p}_i)^\dagger \text{MixedRoPE}(\mathbf{x}_j \mathbf{p}_j). \tag{55}$$

Hence, *any $M$-dimensional LieRE with commutative generators is equivalent to a Mixed RoPE parameterization with learned rotation frequencies.* $\square$

