# OpenReview forum: "A Circular Argument: Does RoPE need to be Equivariant for Vision?"
_NeurIPS.cc/2025/Workshop/UniReps — UniReps2025_

### Official Review · Reviewer_4k1t · 2025-09-04
**Spherical RoPE embeddings -- interesting research direction, preliminary insights**

**Confidence:** 4

**Review:**

This paper introduces spherical RoPE, a multi-dimensional positional embedding method that breaks with equivariance properties of previous approaches by rotating along spherical Euler angles. The authors evaluate spherical RoPE on ImageNet and CIFAR-100 using a supervised ViT-S.
The paper makes a reasonably good effort to introduce and compare different multidimensional RoPE methods. A better exposition of the concepts is difficult with only 4 page extended abstract, Fig 1 helps a lot here. The empirical evidence that equivariance seems to be NOT a crucial property for the success of RoPE is interesting. At this time, the biggest flaw of this work is the evaluation. We see virtually no difference on ImageNet. CIFAR leads to better separation. However, dense prediction tasks such as segmentation would better highlight the importance of the positional embeddings and should be included in the evaluation for the future.

Question to the authors: Do the sin/cos terms in the rotation matrixes create problems with the gradient flow at saddle points for learnable spherical RoPE?

**Score:**

3

**Topic Fit:**

2

---

### Official Review · Reviewer_ndxA · 2025-09-07
**Analyses the importance of equivariance, frequency diversity and oblique directions in (Axial, Mixed) RoPE  for vision tasks.**

**Confidence:** 4

**Review:**

The paper claim 3 hypotheses:
1. Necessity of equivariance in RoPE.
2. Importance of frequency diversity.
3. Significance of oblique direction on performance.

### **Hypotheses #1: Necessity of equivariance in RoPE.**
Outcome: Spherical RoPE was proposed which is non-commutative and non-equivariant. Experiments were conducted on CIFAR100 and ImageNet and Top-1 Accuracy.  \
Spherical RoPE perform at-par when compared to Mixed RoPE, which is equivariant. Based on these results, the paper concludes that RoPE doesn't need to have strict equivariant restriction.

Strength:
1. Spherical RoPE was proposed which with clear explaination of why that's the right choice for ablation of equivariance.
2. Experiments were conducted on smaller subset of the dataset to amplify the importance of equivariance, if any. This didn't change the outcome.

Weakness:
1. Paper mentions that Spherical RoPE is not incapable of learning or approximating equivariance through the network. Any analyses on whether the learned Spherical RoPE network is approximately equivariance or not would be helpful to understand the importance of equivariance.

### **Hypothesis #2: Importance of frequency diversity.**
Outcome: Frequency diversity is important and improves results.

Strength:
1. Uniform RoPE is intuitive and ablated the diverse frequency.
2. Results were backing that.

Weakness:
1. The importance of frequency in text and the role it plays is already proven by "Round and Round We Go! What makes Rotary Positional Encodings useful?" paper, and the finding intuitively translate to Vision. Mentioning this or showing how certain channels in vision model as well are focusing on learning long distance dependency would make the claim stronger.

### **Hypothesis #3: Significance of oblique direction on performance**
Outcome: Oblique directions are not important.

Strength:
1. Proved on the image datasets of relatively small size. Limitation clearly states that this may not translate to all dataset.

Weakness:
1. Oblique directions may not be important for images, but since the paper captures images and videos both, adding the metrics on any one video dataset could be helpful.

### **Overall feedback**
The paper is well-written and is a good contribution in understand the Rotatary position embedding for vision.
Originality: Good. Identifying ways to ablate specific component of RoPE were original and innovative.
Clarity: Excellent. Paper was well-written with appropriate flow and clean articulation.
Significance: Good. It may allow people to apply this to various dataset and conclude whether equivariance is at advantage or disadvantage for certain dataset.
Quality: Excellent.

**Score:**

4

**Topic Fit:**

3

---

### Official Review · Reviewer_gHu4 · 2025-09-10
**Interesting paper**

**Confidence:** 3

**Review:**

# Review

## Summary
This paper presents a theoretical and empirical analysis on which properties of RoPE are important, especially focusing on its shift equivariance and in 2D vision. It is largely composed of two parts - one where a myriad of theoretical results are shared, culminating in many N-D RoPE extensions being equivalent to Mixed RoPE. In the second, more empirical segment, they propose a new positional embedding for images which is not shift equivariant and demonstrate that it works well, casting doubts thus on shift equivariance’s importance.

---

## Strengths

### Originality
- The paper asks an interesting, topical question. There has been much discussion in the last few years on how important symmetries are [EquivScale], although geometric deep learning is often met with the criticism of not being overly relevant. It’s good to apply these questions to relevant settings.
- I especially like proposition 2 as it draws an important conclusion on a range of methods, namely a bunch of RoPE methods for images being equivalent to Mixed RoPE.
- It’s good to study positional encodings in other domains than language, lest something be missed due to overly focusing on a single setting.

### Clarity
- Although the workshop format limits this paper’s readability (as a full conference version likely would as well, as is often so for mathematically inclined works), it does its best to walk the tightrope of showing the high level results and meaning behind them while not being simplistic.
- Although it’s difficult, the very technical and mathematically involved analysis tries being approachable.

### Technical soundness
- I appreciate the paper’s scientific precision, where it makes interesting high level statements that are still very precise and hence quantifiable, e.g. how some kinds of positional encodings affect how models handle oblique features in an image.
- The paper is very principled with its definitions of the different properties, discussing when they are implemented, what they formally mean, and so on. It’s a shame this had to be delegated to the Appendix.
- The Appendix is very thorough.
- The proofs are well laid out and seem straightforward and correct, although I haven’t checked them in detail.

### Significance
- It’s always good to test fundamental assumptions, especially if done rigorously.
- Even just the definitions and proofs are a nice bridge between different proposed methods.

---

## Weaknesses

### Originality
- I don’t see any significant originality weaknesses, in the sense that the paper discusses work that, from some searching and my own domain knowledge, doesn’t seem to have been studied much before in this setting. Its main limitations stem from its significance (below).
- Minor: isn’t Theorem 1 well known, where any high dimensional rotation is decomposable into sets of 2D rotations? Doesn’t this require $n^2$ rotations? Worth clarifying the novelty, whether it’s applicability here or something else, and citing its source if it’s known.

### Clarity
- In the intro the questions and contributions are mixed, making it harder separating the forest from the trees. I recommend separating the questions and motivation from their answers and the work’s contributions.
- Some notation isn’t immediately clear when this is presented in a workshop format, e.g. the meaning of the different symbols in Eq. 2, like *p*. Especially for people less familiar with positional encodings, this should be improved. It would be good showing RoPE’s typical formulation as well.
- The two parts, theory and experiments, currently feel disconnected. There’s some motivation linking them which can be made clearer, currently the flow feels off.
- There are many specific places where clarity can be improved. For example, in lines 23-24 *“make constraints on the rotations”* is unclear for someone without a lot of context.
- Motivation for introducing some definitions can be better, e.g. linear flows.

### Technical soundness
- Especially as typical images likely don’t require long contexts it’s unclear how well these results generalize. It would be interesting to test them in more equivalent settings, e.g. high resolution images and ViTs with small patches. Wouldn’t it make sense that equivariance is less important for a smaller domain? This should be further discussed and emphasised, at least to clarify it, or shown more thoroughly empirically.
- Deep learning is a leaky abstraction (see Karpathy’s blog post on debugging neural networks), so it’s easy for different mechanisms to cover for one’s shortcoming. It would be good to isolate these via ablations so you can see how important different assumptions and design choices are.
  - As it is, some things are stated but not thoroughly shown. For example, does having positional encodings with various properties make the model more robust to some transformations, like rotations? This would show whether designing positional encodings that better model oblique angles has a big impact or if the model can anyway learn to handle them well.
- To answer the basic question of *“is equivariance important for PEs”* aren’t there much simpler ways to achieve symmetry breaking than thinking about generators/commutators/etc.? That approach doesn’t feel well motivated, or if there is a good motivation for it over simpler methods then it’s currently missing.
- In spite of their mentioned limitations, modern SOTA ViTs (e.g. DINOv3) use axial RoPE [DINOv3]. It would be interesting to theorise why they don’t see mentioned shortcomings, or much more impressively demonstrate these shortcomings on pretrained open source foundation models.
- Doesn’t it make sense that equivariant PEs are less important for typical image distributions? For example, the sky is often in the top half of a picture. It would be good to at least discuss this, if not ablate. This is unlike language where very similar sentences can easily be shifted across parts of a document.

### Significance
- I recommend discussing *why* one would want to test these assumptions, not only the tests themselves.
  - There are many assumptions we make in deep learning, being an empirical engineering science, but we usually test something if we think it might be wrong.
  - For example, some works show that equivariance might harm optimisation [REMUL] or not be necessary at scale [EquivScale], this could be an instantiation thereof.
- Why specifically study this for vision? Why not have it be a general analysis, applicable to 1D modelling like language as well? As the main important use of PEs is in language modelling, this limits this work’s applicability quite a bit. The basic question of whether shift equivariance is important can equally well be asked there.
- Although applications to many modalities in different numbers of dimensions are mentioned, only 2D vision is studied.

---

## Questions / Suggestions / Misc
- If different PE schemes like LieRE are more general than RoPE why aren’t they used? It’s worth understanding this to see if using Axial RoPE has any real limitations, e.g. do mentioned shortcomings disappear at scale? Even if this can’t be tested using academic compute, it should be discussed.
- I recommend running the experiments with fewer data augmentations, they can decrease reliance on good inductive biases via PEs. E.g. if you have a model that gets all permutations of a set of numbers it’ll learn to be more permutation invariant than a model that only gets the numbers as they are.
- Wouldn’t it make sense to test capacity-limited models, so they have to rely on the inductive biases more? Would think capacity matters more than data here.
- For targeted ablations synthetic data might work well. See CoordConv for some potential inspiration [CoordConv].
- *“However, while RoPE is often claimed to be successful due to its shift-equivariance, the validity of that claim and necessity of equivariance has not been thoroughly tested.”* - the RoPE paper mentions a few different upsides of RoPE, this is just one of them, worth being more precise.
- *“One rotate the same pair by both positional coordinates where the amount of rotation caused by each is a parameterized for each pair”* — what?
- `384, eq ??`
- `441-443` — is unclear
- `507-508` — should be made clearer if this is a definition, handwaving is fine but make it clear which it is
- `523` — how does x=0 result in one here, not in the exponent?
- `546` — would be nice explaining why operator variance is a natural way to get locality. Unsure I get the naturalness point, why is this a better way to measure spread than anything else?
- `549-551` — how does Barbero et al preclude RoPE being relative? Do you mean local? Don’t think it shows anything against locality.
- `566, eq ??`
- `54-56` — why would we want to extend it to M-D?
- Eq2 — is this for all d all lambda x are zero or for every d either lambda x or lambda y is zero?
- *Affect*, not *effect* in Fig. 1 caption.
- Why does >1D RoPE require giving up equivariance/relativity? On that note, recommend using more consistent terms.
- `85-90` — what *“this”* refers to is unclear.

---

## Overall
To summarize, the paper has many interesting points but could be made tighter. Its main limitation is its potentially limited applicability, which might seem less than it is from some current clarity issues.

If it’s aiming to be less applicable but moreso give theoretical insights, then that should be emphasized.

#### References

[DINOv3] Siméoni, Oriane, et al. "DINOv3." arXiv preprint arXiv:2508.10104 (2025).

[EquivScale] Brehmer, Johann, et al. "Does equivariance matter at scale?." arXiv preprint arXiv:2410.23179 (2024).

[CoordConv] Liu, Rosanne, et al. "An intriguing failing of convolutional neural networks and the coordconv solution." Advances in neural information processing systems 31 (2018).

[REMUL] Elhag, Ahmed A., et al. "Relaxed Equivariance via Multitask Learning." arXiv preprint arXiv:2410.17878 (2024).

**Score:**

3

**Topic Fit:**

3

---

### Official Review · Reviewer_MWZj · 2025-09-15

**Confidence:** 3

**Review:**

Strengths
- **Novelty on RoPE**: The paper introduces Spherical RoPE, which applies spherical rotations to embedding vectors in order to investigate whether non-commutative rotations can benefit vision tasks.
- **Mathematical rigor**: Proofs are presented with solid mathematical rigor, particularly in describing the separability and locality of RoPE. This theoretical grounding provides a reasonable motivation for the proposed work.
- **Performance**: Spherical RoPE shows stronger results than other embeddings in classification and segmentation tasks. The paper also draws thoughtful and meaningful connections between LieRE and vision embeddings.

Weaknesses
- **Unsupported claim**: The paper challenges the importance of equivariance in RoPE, but experiments provided are too limited in scope to support this claim. The study would benefit from further ablations of existing RoPE methods.
- **Limited theoretical novelty**: Apart from heavy discussion of previous works, the primary contribution lies in embedding images into a spherical domain. However, closely related works such as 3D-RPE, VRoPE, and VideoRoPE are not discussed or compared against.
- **Evaluation concerns**: (1) Experiments are limited to classification and segmentation, without extending to more complex tasks such as visual reasoning. (2) Gains from Spherical RoPE appear modest compared to Mixed RoPE. (3) No evaluation on video embeddings is included, despite the natural extension to this setting.

Additional comments
- line 51, 55, 57, 58: Notations $N$, $M$, $D$, $A_x$, $A_y$, $p_x$, $p_y$ are unclear.
- line 566: Reference error with "Eq. ??".
- Overall writing would benefit from reorganization and clearer exposition.
- Potential complexity overhead of Spherical RoPE relative to existing methods deserves discussion.

**Quality:** 2: fair

**Clarity:** 2: fair

**Originality:** 1: poor

**Significance:** 2: fair

**Score:**

2

**Topic Fit:**

1